# Epigenetic Deregulation of Protein Tyrosine Kinase 6 Promotes Carcinogenesis of Oral Squamous Cell Carcinoma

**DOI:** 10.3390/ijms23094495

**Published:** 2022-04-19

**Authors:** Yi-Ping Hsieh, Ken-Chung Chen, Meng-Yen Chen, Ling-Yu Huang, An-Yu Su, Wei-Fan Chiang, Wen-Tsung Huang, Tze-Ta Huang

**Affiliations:** 1Institute of Basic Medical Sciences, College of Medicine, National Cheng Kung University, Tainan 701401, Taiwan; pin1000822@gmail.com; 2Institute of Oral Medicine, Department of Dentistry, College of Medicine, National Cheng Kung University, Tainan 701401, Taiwan; omsboy@gmail.com (K.-C.C.); ccdc0002.tw@gmail.com (M.-Y.C.); 3Division of Oral and Maxillofacial Surgery, Department of Stomatology, National Cheng Kung University Hospital, Tainan 701401, Taiwan; 4Institute of Clinical Medicine, College of Medicine, National Cheng Kung University, Tainan 701401, Taiwan; helen85116@gmail.com; 5Department of Dentistry, College of Medicine, National Cheng Kung University, Tainan 701401, Taiwan; erinanyusu@gmail.com; 6Chi Mei Medical Center, Liouying, Tainan 72263, Taiwan; bigfanfan@yahoo.com.tw; 7School of Dentistry, National Yang Ming University, Taipei 11221, Taiwan

**Keywords:** oral squamous cell carcinoma, oral squamous cell carcinoma carcinogenesis, epigenetic regulation, protein tyrosine kinase 6 (PTK6)

## Abstract

Oral squamous cell carcinoma (OSCC) accounts for over 90% of oral cancers and causes considerable morbidity and mortality. Epigenetic deregulation is a common mechanism underlying carcinogenesis. DNA methylation deregulation is the epigenetic change observed during the transformation of normal cells to precancerous and eventually cancer cells. This study investigated the DNA methylation patterns of *PTK6* during the development of OSCC. Bisulfite genomic DNA sequencing was performed to determine the *PTK6* methylation level. OSCC animal models were established to examine changes in PTK6 expression in the different stages of OSCC development. The DNA methylation of *PTK6* was decreased during the development of OSCC. The mRNA and protein expression of PTK6 was increased in OSCC cell lines compared with human normal oral keratinocytes. In mice, the methylation level of *PTK6* decreased after treatment with 4-nitroquinoline 1-oxide and arecoline, and the mRNA and protein expression of PTK6 was increased. *PTK6* hypomethylation can be a diagnostic marker of OSCC. Upregulation of PTK6 promoted the proliferation, migration, and invasion of OSCC cells. PTK6 promoted carcinogenesis and metastasis by increasing STAT3 phosphorylation and ZEB1 expression. The epigenetic deregulation of *PTK6* can serve as a biomarker for the early detection of OSCC and as a treatment target.

## 1. Introduction

Oral squamous cell carcinoma (OSCC) is the most common form of head and neck cancer (HNSC), and the average 5-year survival rate of patients with advanced OSCC is approximately 40% [1]. Tobacco smoking, alcohol consumption, and betel quid chewing are the main etiological risk factors for OSCC [2]. Surgery, chemotherapy, radiotherapy, and immune therapy, separately or in combination, can be used to treat OSCC, depending on the clinical stage [3]. OSCC is characterized by aggressive biological behavior and regional metastasis. Lymph node metastasis is a critical prognostic indicator for OSCC and it reduces the survival rate by 50%. Early detection of OSCC improves prognosis, with the survival rate of patients with localized OSCC reaching up to 82%. Delayed diagnosis results in high morbidity and mortality [4,5]. However, except for direct mucosal inspection by a physician, reliable biomarkers for the early detection of OSCC are not yet available.

The carcinogenesis of OSCC occurs gradually and is preceded by the development of oral potentially malignant disorders (OPMDs), including oral leukoplakia, erythroplakia, erythroleukoplakia, and oral submucous fibrosis, which may eventually progress to oral cancer [6,7]. Compared with healthy individuals, patients with OPMDs have a higher risk of OSCC [8]. However, the diagnosis is subjective because of the difficulty in predicting whether lesions are likely to undergo malignant transformation. Therefore, biomarkers for predicting the malignant potential of premalignant lesions should be urgently identified.

Epigenetic regulation involves the modulation of gene expression states without the modification of DNA sequences, and epigenetic effects are long-lasting and reversible [9]. Epigenetic mechanisms including DNA methylation, histone modification, chromatin condensation, and micro-RNA regulation maintain normal growth, development, and gene expression in different organs [10]. Increasing evidence suggests that epigenetic aberrations in cancer cells are primarily based on initial events associated with carcinogenesis [11]. DNA methylation is a biological process that transfers a methyl group from S-adenosyl methionine to the fifth carbon of a cytosine [12]. Genomic methylation patterns are frequently altered in many cancers that involve global hypomethylation and regional hypermethylation. Global hypomethylation induces genomic instability that contributes to cell transformation [13]. Hypermethylation and hypomethylation of the same CpG islands are observed in different tumors, highlighting the dynamics of epigenetic plasticity during cancer development [14].

Protein tyrosine kinase 6 (PTK6), also known as breast tumor kinase, is an intracellular tyrosine kinase. The PTK6 protein structure is closely related to that of Src tyrosine kinase, possessing SH2 and SH3 domains, but it lacks a membrane-targeting myristoylation signal [15]. PTK6 is highly expressed and activated in multiple cancer types, including breast, prostate, pancreatic, and oral cancers [16,17,18]. Emerging evidence indicates that PTK6 regulates oncogenic processes such as cell proliferation, invasion, and migration [19,20,21]. In breast cancer, PTK6 has also been verified to play a critical role in epithelial–mesenchymal transition (EMT). PTK6 cooperates with HER2 and Src to regulate EMT in HER2-positive breast cancer cells. PTK6 suppresses E-cadherin expression and increases mesenchymal markers in nontransformed breast epithelial cells, to promote cell migration [22,23]. Moreover, a high expression of PTK6 in nondividing epithelial cells was observed to suppress cell proliferation and promote cell differentiation and apoptosis [24]. Such dual functions of PTK6 might be associated with its subcellular localization [25]. PTK6 localized at the nucleus phosphorylates the RNA-binding protein Sam68, which suppresses PTK6-induced cell proliferation [26]. In contrast, cytoplasmic PTK6 phosphorylates many intracellular targets to promote oncogenic functions, including p38/MAPK and JAK2/STAT [27,28]. STAT3 is a substrate of PTK6 in the cytoplasm. In colon cancer, PTK6 interacts with JAK2 and increases STAT3 phosphorylation to enhance stemness and chemoresistance. PTK6 also promotes STAT3 and ERK5 activation to promote cell survival [28]. However, the mechanisms underlying the deregulation of PTK6 in OSCC remain unclear.

This study investigated the DNA methylation patterns of *PTK6* during the development of OSCC. The results of this study revealed that the DNA methylation of *PTK6* decreased with the progression of OSCC. High expression of PTK6 facilitated OSCC cell proliferation, migration, and invasion. In addition, we identified PTK6 oncogenic pathways that promote cell proliferation and migration by increasing STAT3 phosphorylation and ZEB1 expression. Our results indicated that the epigenetic deregulation of *PTK6* can serve as a biomarker for the early detection of OSCC and as a treatment target.

## 2. Results

### 2.1. PTK6 Methylation Deregulation Induces OSCC Carcinogenesis

In our previous study, we investigated global differential methylation in the normal oral mucosa and in OSCC specimens with or without neck lymph node metastasis using methyl-CpG binding-domain–based capture followed by high-throughput sequencing. We identified 1462 differentially methylated CpG islands in the OSCC specimens compared with normal controls and observed that 359 loci were linked to lymph node metastasis. The DNA methylation level of PTK6 was changed in the OSCC specimens compared with the normal oral mucosa. Moreover, the DNA methylation of PTK6 was considerably decreased in the lymph node metastasis tissue samples [29].To determine the relationship between the PTK6 methylation level and OSCC progression, we examined the PTK6 methylation level in normal tissue, precancerous lesions, and OSCC samples, reflecting the different stages of OSCC development. We identified two specific loci on the PTK6 exon 1 and analyzed the methylation status of CG sites (Figure 1a). The results revealed that the PTK6 methylation of specific region 1 or specific region 2 was significantly decreased during OSCC development (Figure 1b). The methylation of specific region 1 was decreased in most of the OSCC specimens compared with their corresponding normal tissues (Figure 1c). In addition, we examined the DNA methylation level of PTK6 in the saliva samples of patients with OSCC. The results demonstrated that the PTK6 methylation of specific region 1 and specific region 2 was decreased in the saliva samples (Figure 1d). Methylation data from The Cancer Genome Atlas (TCGA) database were used to confirm changes in the methylation of PTK6. The data indicated that PTK6 hypomethylation can be used as a diagnostic marker of HNSC (Appendix A).

The transcription and translation levels of PTK6 were validated in human normal oral keratinocytes (HNOKs) and OSCC cell lines. The results indicated that the methylation of PTK6 was decreased and that the mRNA and protein expression of PTK6 was increased in the OSCC cell lines compared with HNOKs. The methylation of PTK6 was negatively correlated with its transcription level (Appendix A). These results revealed that the hypomethylation of PTK6 may be a critical mechanism, resulting in the deregulation of PTK6 in OSCC carcinogenesis.

An animal model of OSCC carcinogenesis was established to confirm the correlation of the DNA methylation of PTK6 with its mRNA transcription in vivo. Male C57BL/6J mice were treated with 4-nitroquinoline 1-oxide (4-NQO) and arecoline to induce the development of OSCC. After treatment of the normal oral mucosa with 4-NQO and arecoline for a certain period, the gradual development of OPMDs and eventually OSCC was observed. The mice were sacrificed at 8, 16, 20, and 29 weeks, and changes were examined in the different stages of OSCC development (Figure 2a). Subsequently, we analyzed the PTK6 DNA methylation level in the tongue tissues of the mice treated with 4-NQO and arecoline at 8, 16, 20, and 29 weeks. After treatment with 4-NQO and arecoline, the methylation level of PTK6 decreased with the progression of OSCC development in the patient samples (Figure 2b). The mRNA and protein expression of PTK6 increased with time in the mouse tongue samples (Figure 2c,d). These results demonstrated that PTK6 methylation deregulation induces OSCC carcinogenesis by increasing PTK6 expression.

### 2.2. Upregulation of PTK6 Promotes OSCC Cell Proliferation, Migration, and Invasion

To explore the oncogenic functions of PTK6 in OSCC, OCMC and OECM1 cell lines were transfected with short hairpin RNA (shRNA) to knock down PTK6 expression. The expression of PTK6 in the cells was confirmed through Western blotting and quantitative polymerase chain reaction (qPCR; Figure 3a). To investigate the effect of PTK6 on the proliferation ability of the OSCC cells, we performed a WST-1 assay. The PTK6 knockdown cells did not exhibit significant differences in cell proliferation (Figure 3b). The effect of PTK6 on the metastatic potential in vitro was examined using transwell migration and invasion assays. The results revealed that PTK6 downregulation significantly reduced the migration and invasion abilities of the OSCC cells (Figure 3c,d). PTK6 overexpression was induced in the OML3 and OCSL cells, and PTK6 expression in the cells was confirmed through Western blotting and qPCR (Figure 4a). PTK6 overexpression significantly increased the proliferation rate of the OSCC cells, as observed in the WST-1 assay (Figure 4b). Moreover, the upregulation of PTK6 promoted the migration and invasion of the OSCC cells (Figure 4c,d).

To verify the role of PTK6 in OSCC progression in vivo, the control cells and PTK6 knockdown and overexpression cells were orthotopically injected into the right buccal mucosa of nude mice to establish the xenograft tumor model. The results revealed the effective inhibition of tumor growth and marked reductions in tumor volume and weight following PTK6 knockdown (Figure 5a–c). The overexpression of PTK6 significantly increased tumor volume and weight (Figure 5d–f). In summary, these data indicated that PTK6 plays a crucial role in promoting the migration, invasion, and carcinogenesis of the OSCC cells.

### 2.3. PTK6 Promoted Carcinogenesis and Metastasis through STAT3 and ZEB1

The mechanisms through which PTK6 promotes OSCC cell migration, invasion, and carcinogenesis were analyzed through Western blotting downstream of PTK6. Epithelial–mesenchymal transition (EMT) is an essential process that promotes tumor metastasis [30]. The expression of the epithelial cell marker E-cadherin, the mesenchymal cell marker N-cadherin, and EMT-related transcription factors was evaluated in PTK6 knockdown and overexpression cells. The results revealed that the expression of N-cadherin and ZEB1 was decreased in PTK6 knockdown cells but increased in PTK6 overexpression cells. The expression of E-cadherin, Snail, and Slug did not significantly change in PTK6 knockdown and overexpression cells. STAT3 is necessary for cell proliferation and survival in many cancers, including OSCC [31,32]. Next, we investigated whether STAT3, downstream of PTK6, promotes OSCC carcinogenesis. The results revealed that STAT3 phosphorylation was decreased in PTK6 knockdown cells, whereas STAT3 phosphorylation was increased in PTK6 overexpression cells (Figure 6). These findings indicated that PTK6 promoted OSCC metastasis and carcinogenesis by increasing ZEB1 expression and STAT3 phosphorylation.

## 3. Discussion

PTK6 upregulation has been identified in multiple cancer types and is associated with poor patient prognosis [18,33,34,35]. Previous research investigated the expression of PTK6 in human oral squamous cell carcinomas and normal oral epithelium (NOE). The results showed that NOE expresses higher levels of PTK6 compared with OSCC cells [36]. However, the mechanisms underlying PTK6 dysregulation in OSCC remain unclear. Previous studies indicated that the methylation densities in promoters and first exons were negatively correlated with the corresponding gene expression level [37]. Two specific loci on the *PTK6* exon 1 were identified in this research, and the methylation status of CG sites analyzed. The fact that *PTK6* methylation of specific region 1 or specific region 2 was gradually decreased during the carcinogenesis of OSCC was also validated. The results were confirmed in an animal model treated with 4-NQO and arecoline, which exhibited a decrease in the methylation level of *PTK6* with time. Subsequently, the mRNA and protein expression of PTK6 increased, following treatment with 4-NQO and arecoline. The methylation of *PTK6* was decreased and the mRNA and protein expression of PTK6 was increased in the OSCC cells compared with HNOKs. Therefore, the methylation of *PTK6* was negatively correlated with its transcription level. We determined that *PTK6* hypomethylation increased the PTK6 transcription level in OSCC progression. The main risk factor for OSCC in Taiwan is betel quid chewing [38]. However, HPV infection and the use of tobacco and alcohol and are the main etiological risk factors for OSCC in the USA [39,40,41]. Different risk factors for inducing OSCC carcinogenesis may involve different mechanisms. Therefore, the PTK6 expression in OSCC is worth exploring in detail in different countries.

PTK6 is involved in diverse cellular processes, including cell proliferation, migration, invasion, and apoptosis inhibition. PTK6 and extracellular signal-regulated kinase 5 (ERK5) mediate Met receptor signaling to promote breast cancer cell migration [42]. PTK6 regulates insulin-like growth factor to enhance the anchorage-independent survival of breast cancer cells [43]. Furthermore, PTK6 inhibits the apoptosis of lapatinib-resistant Her2-positive breast cancer cells by suppressing Bim expression through p38 inactivation [44]. Our results indicated that the upregulation of PTK6 increased OSCC cell proliferation, migration, invasion, and carcinogenesis.

PTK6 plays a critical role in EMT. PTK6 cooperates with HER2 and Src to regulate cell survival and EMT in HER2-positive breast cancer cells [22]. PTK6 suppresses E-cadherin expression and increases mesenchymal markers in nontransformed breast epithelial cells, to promote cell migration [23]. EMT is a critical process that promotes the migration and invasion abilities of epithelial cells [45]. The expression of the epithelial cell marker E-cadherin, the mesenchymal cell marker N-cadherin, and EMT-related transcription factors was examined in PTK6 knockdown and overexpression cells. The results revealed that the expression of N-cadherin and ZEB1 was decreased in PTK6 knockdown cells but increased in PTK6 overexpression cells. PTK6 promoted the migration and invasion of the OSCC cells by increasing the expression of N-cadherin and ZEB1. STAT3 is a well-known substrate of PTK6. In colon cancer, PTK6 interacts with JAK2 and increases STAT3 phosphorylation to enhance stemness and chemoresistance [28]. PTK6 promotes STAT3 and ERK5 activation to promote cell survival and response to DNA-damaging treatments in colon cancer cells [46]. STAT3 is required for proliferation in multiple cancer types including glioblastoma, colorectal cancer, and OSCC [31,47,48]. Thus, we investigated whether PTK6 promotes the proliferation of OSCC cells. Our data revealed that STAT3 phosphorylation was decreased in PTK6 knockdown cells but increased in PTK6 overexpression cells. PTK6 promoted the proliferation of the OSCC cells by increasing STAT3 phosphorylation.

In conclusion, this study demonstrated that the DNA methylation of *PTK6* was decreased during the carcinogenesis of OSCC. The methylation level of *PTK6* was negatively correlated with its transcription level. Upregulation of PTK6 facilitated the proliferation, migration, invasion, and carcinogenesis of the OSCC cells. ZEB1 and STAT3, which are downstream of PTK6, promote OSCC metastasis and proliferation. The epigenetic deregulation of *PTK6* can serve as a biomarker for the early detection of OSCC and as a treatment target.

## 4. Materials and Methods

### 4.1. Cell Lines and Culture

Immortalized HNOKs (human normal oral keratinocytes cell line) were received from the laboratory of Professor Dar-Bin Shieh. The human OSCC cell lines OCMC, OECM, OC2, OML3, OCSL, SAS, and SCC-25 were received from the laboratory of Professor Kuo-Wei Chang. HNOKs were cultured in keratinocyte medium containing human recombinant epidermal growth factor and bovine pituitary extract (Gibco, New York, NY, USA). OCMC, OECM, OC2, OML3, and OCSL cells were cultured in PRMI1640 medium (Gibco, New York, NY, USA). SAS cells were cultured in Dulbecco’s Modified Eagle Medium (DMEM) (Gibco, New York, NY, USA). SCC-25 cells were cultured in DMEM/F12 medium (Gibco, New York, NY, USA). The cell culture media contained 10% fetal bovine serum (Gibco, New York, NY, USA), 1% L-glutamine (Simply, Taoyuan, Taiwan), and 1% antibiotic–antimycotic (Simply, Taoyuan, Taiwan). The cells were maintained in a humidified chamber under 5% CO_2_ in air at 37 °C until they were used in experiments.

### 4.2. Clinical Patient Specimen Collection

This research was approved by the Institutional Review Board of National Cheng Kung University Hospital (IRB No: B-ER-103-072) and complied with medical research protocols established in the Declaration of Helsinki by the World Medical Association. Subjects. The patients who were scheduled for surgery according to therapy needs were recruited. After signed the inform consent, normal tissue, OPMD lesions, and OSCC tissue were collected after patients finished the treatment. Normal and OSCC patients’ saliva samples were collected before patients were scheduled for therapy. In total, 35 normal tissues, 18 precancerous lesions, and 51 OSCC patient samples were recruited for PTK6 methylation analysis. In addition, 10 normal and 10 OSCC patients’ saliva samples were also recruited for PTK6 methylation analysis.

### 4.3. OSCC Animal Models

Animal experiments were approved by the Animal Care and Use Committee (IACUC) of National Cheng Kung University. Male C57BL/6J mice raised on the Tainan campus of the National Institutes of Health were used in this study. The mice were randomly divided into the control and experimental groups. The control group was given normal drinking water. The experimental group was given drinking water containing 4-NQO (200 μg/mL; Sigma-Aldrich, Saint Louis, MO, USA) and arecoline (200 μg/mL; TCI, Tokyo, Japan) to induce the development of OSCC. The mice in the control and experimental groups were sacrificed at 8, 16, 20, and 29 weeks and the different stages of OSCC development were determined. There were 3 control and 7 4-NQO-and-arecoline-treated mice in the 8 weeks group, 3 control and 4 4-NQO-and-arecoline-treated mice in the 16 weeks group, 3 control and 6 4-NQO-and-arecoline-treated mice in the 20 weeks group, and 3 control and 4 4-NQO-and-arecoline-treated mice in the 29 weeks group. About 3x12 mm of tissue was cut from each mouse, and about 150 µg of protein lysate was obtained. Fifty micrograms of total protein lysate was used for the Western blotting analysis. DNA and RNA were extracted from the tongue tissues of the mice to examine changes in PTK6 expression in the different stages of OSCC development.

### 4.4. PTK6 Knockdown and Overexpression in Cell Lines

The shRNA specifically targeting PTK6 and the pLKO.1-emptyT vector were obtained from the National RNAi Core of Academia Sinica (clone ID TRCN0000021551, Taipei, Taiwan). A total of 10 µg of pLKO.1Ppuro and 10 µg of pLKO.1-shPTK6 plasmids were transfected into HEK293T cells with Lipofectamine™ 3000 (Invitrogen, Waltham, MA, USA) and produced lentiviruses, respectively. The lentiviruses were used to infect OSCC cell lines (OCMC and OECM1) expressing a high level of PTK6 (multiplicity of infection, MOI = 3). The empty vector pLAS2W-Puro was obtained from the National RNAi Core of Academia Sinica. The pLAS2W-PTK6 expression plasmid was synthesized by Protech Technology Enterprise. A total of 10 µg of pLAS2W-Puro and 10 µg of pLAS2W-PTK6 expression plasmids were transfected into HEK293T cell lines with Lipofectamine™ 3000 and produced lentiviruses, respectively. The lentiviruses were used to infect OSCC cell lines (OML3 and OCSL) expressing a low level of PTK6 (MOI = 3). Subsequently, 0.5 µg/mL of puromycin was used to select vector control and PTK6 knockdown and overexpression cells. After treatment with puromycin for 3 days, real-time reverse transcription polymerase chain reaction and Western blotting were used to analyze PTK6 mRNA and protein expression. The OSCC PTK6 knockdown or overexpression cells were used to analyze cell functions.

### 4.5. qPCR

qPCR was used to confirm the knockdown and overexpression of PTK6 in the OSCC cell lines. Total RNA was purified using TRI reagent (Zymo Research, Irvine, CA, USA) and Direct-zol RNA kits (Zymo Research, Irvine, CA, USA), according to the manufacturer’s instructions. The concentration and quality of the extracted RNA were examined using NanoDrop. Complementary DNA (cDNA) was synthesized using a qPCRBIO cDNA synthesis kit (PCR Biosystems, London, UK). The relative quantity of mRNA was analyzed using a qPCRBIO SyGreen Mix (PCR Biosystems, London, UK) and the StepOnePlus Real-Time PCR System (Applied Biosystems, London, UK). The expression levels of target genes were quantified using the comparative ΔCT method and normalized to the expression level of GAPDH mRNA. The following primer sequences were used: PTK6 qPCR (forward, GGCTATGTGCCCACAACTAC; reverse, GGGAGATGCAGCCAAAGAAC) and GAPDH qPCR (forward, CCCCTTCATTGACCTCAACTACAT; reverse, CGCTCCTGGAAGATGGTGA).

### 4.6. Western Blotting

Western blotting was performed to investigate the expression of genes downstream of PTK6. Total cellular proteins were extracted using RIPA lysis buffer and a protease inhibitor cocktail (Sigma-Aldrich, Saint Louis, MO, USA). Then, the protein concentration was quantified using a Bio-Rad protein assay kit (Bio-Rad, Hercules, CA, USA). Protein lysates were separated using sodium dodecyl sulfate–polyacrylamide gel electrophoresis and electroblotted onto polyvinylidene fluoride membranes (Millipore, Burlington, MA, USA). Then, the membranes were incubated with a primary antibody, followed by a horseradish peroxidase (HRP)-conjugated secondary antibody. The protein bands were detected using a chemiluminescent HRP substrate (Millipore, Burlington, MA, USA). The relative expression level of proteins was measured using ImageJ software. GAPDH was used as the internal control. The following antibodies were used: human PTK6 (1:1000 dilution) (sc-166171, Santa Cruz Biotechnology, Dallas, TX, USA), human p-STAT3 (1:1000 dilution) (9131, Cell signaling, Danvers, MA, USA), human E-cadherin (1:1000 dilution) (20874-1-AP, Proteintech, Chicago, IL, USA), human N-cadherin (1:1000 dilution) (22018-1-AP, Proteintech, Chicago, IL, USA), human Snail (1:1000 dilution) (A5243, Abclonal, Woburn, MA, USA), human Slug (1:1000 dilution) (12129-1-AP, Proteintech, Chicago, IL, USA), human ZEB1 (1:1000 dilution) (A16981, Abclonal, Woburn, MA, USA), and human GAPDH (1:1000 dilution) (G8795, Sigma-Aldrich, Saint Louis, MO, USA).

### 4.7. Bisulfite Genomic DNA Sequencing

Genomic DNA isolated from the OSCC cell lines and patient samples was purified using a DNA isolation kit (Zymo Research, Irvine, CA, USA) and treated with sodium bisulfite using an EZ DNA methylation kit (Zymo Research, Irvine, CA, USA). Subsequently, a polymerase chain reaction (PCR) was performed to amplify the *PTK6* promoter region, using Taq DNA polymerase (Invitrogen, Waltham, MA, USA). The pyrosequencing of the PCR products was performed using a PyroMark Gold Q24 Reagent kit (Qiagen, Hilden, Germany) to analyze the methylation status of the *PTK6* promoter. Primers and PCR conditions designed using PyroMark Assay Design 2.0 software were used to amplify specific CpG islands. Methylated CpG sites were detected using a PyroMark Q24 system. The PTK6 S1 sequence for detection was CCGGACGGACGAGGAGCTGAGCTTCCGCGCGGGGGACGTCTTCCACGTGGCC (+125~+171) and the pyrosequence-designed nucleotide sequence for analysis was TYGGAYGGAYGAGGAGTTGAGTTTTYGYGYGGGGGAYGTTTTTTAYGTGG. The PTK6 S2 sequence for detection was TACCTGGCCGAGAGGGAGACGGTGGAGTCGGAACCGTGCG (+266~+298) and the pyrosequence-designed nucleotide sequence for analysis was TATTTGGTYGAGAGGGAGAYGGTGGAGTYGGAATYGTGYGTGTTTTAGGT. The primer sequences used in PCR and pyrosequencing are shown in Appendix A.

### 4.8. Proliferation Assay

The cell proliferation rate was analyzed using a WST-1 cell proliferation assay kit (TaKaRa, Kusatsu shi, Japan). The cells were seeded at a density of 3000 cells per well in 96-well plates and left to adhere overnight. The cell number was determined using the WST-1 reagent. Absorbance was measured at 450 nm at seven time points (0, 24, 48, 72, 96, 120, and 144 h) to calculate the proliferation rate.

### 4.9. In vitro Transwell Migration and Invasion Assays

A total of 8 × 10^4^ cells were plated in the upper chamber of cell culture inserts (Falcon, 353495) coated with (for the invasion assay) or without (for the migration assay) Matrigel (Corning, Corning, NY, USA) and incubated at 37 °C in 5% CO_2_ for 24 h. The migrated cells retained on the bottom surface of the membrane were fixed and stained with trypan blue solution (Merck, Branchburg, NJ, USA). The nonmigrated cells were scraped from the upper surface of the membrane using a cotton swab. The cells were photographed and quantified using ImageJ software.

### 4.10. Xenograft Tumor Model

Animal experiments were approved by the IACUC of National Cheng Kung University. Five-week-old male nude mice (CAnN.Cg-Foxn1nu/CrlNarl) were purchased from the National Laboratory Animal Center. A total of 1 × 10^6^ cells were orthotopically injected into the right buccal mucosa of the nude mice. The tumor sizes and weights were measured every two days. The tumor volumes and weights were examined after sacrificing the mice. Tumor volume was calculated as follows: tumor volume (mm^3^) = (length × weight^2^)/2.

### 4.11. Statistical Analysis

Data are expressed as the mean ± standard deviation. All statistical analyses were performed using GraphPad Prism statistical software. Statistical significance was set at * *p* < 0.05, ** *p* < 0.01, and *** *p* < 0.005 and examined using a two-tailed Student’s *t*-test.

## Figures and Tables

**Figure 1 ijms-23-04495-f001:**
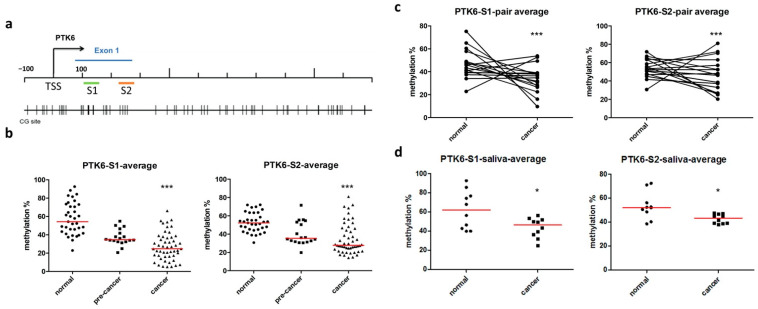
PTK6 was hypomethylated with OSCC development. (**a**) CpG sites on the PTK6 exon 1 and two specific regions were designed for pyrosequencing. (**b**) The methylation levels of PTK6 in 35 normal tissues, 18 precancerous lesions, and 51 OSCC patient samples were measured through bisulfite pyrosequencing. (**c**) The relative DNA methylation of PTK6 in tumor tissues and their corresponding normal tissues from the same patient. (**d**) The methylation levels of PTK6 in 10 normal and 10 OSCC patients’ saliva samples were measured through bisulfite pyrosequencing. * *p* < 0.05. *** *p* < 0.005.

**Figure 2 ijms-23-04495-f002:**
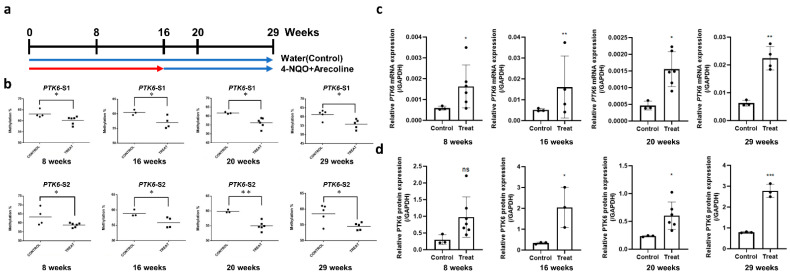
PTK6 hypomethylation induces OSCC carcinogenesis by increasing PTK6 transcriptional expression in the animal model. (**a**) Flowchart of exposure of mice to carcinogens (200 µg/mL of 4-NQO and 500 µg/mL of arecoline) in drinking water. The mice treated with 4-NQO and arecoline and corresponding control mice were sacrificed at 8, 16, 20, and 29 weeks for analysis. (**b**) Methylation analysis of PTK6 in the tongue tissues of OSCC mice at different time points. (**c**) PTK6 transcriptional expression in the tongue tissues of the OSCC mice at different time points. (**d**) The protein expression of PTK6 in the tongue tissues of the OSCC mice at different time points. * *p* < 0.05. ** *p* < 0.01; *** *p* < 0.005.

**Figure 3 ijms-23-04495-f003:**
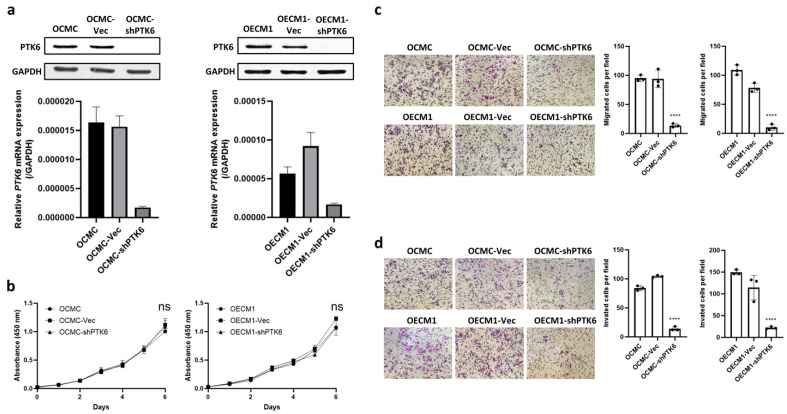
Downregulation of PTK6 suppressed OSCC cell migration and invasion. (**a**) OCMC and OECM1 cell lines were used to knock down PTK6 by using shRNA. The mRNA and protein expression were examined through real-time PCR and Western blotting in PTK6 knockdown cells. (**b**) Cell viability of control and PTK6 knockdown cells was evaluated using a WST-1 assay. Absorbance was measured at 450 nm at seven time points (0, 24, 48, 72, 96, 120, and 144 h) to calculate the proliferation rate. (**c**) The migration abilities of control and PTK6 knockdown cells were examined using a transwell assay. Migrated cells were quantified using ImageJ. (**d**) The invasive abilities of control and PTK6 knockdown cells were measured using a transwell Matrigel assay. Invasive cells were quantified using ImageJ. ns means not significant. **** *p* < 0.001.

**Figure 4 ijms-23-04495-f004:**
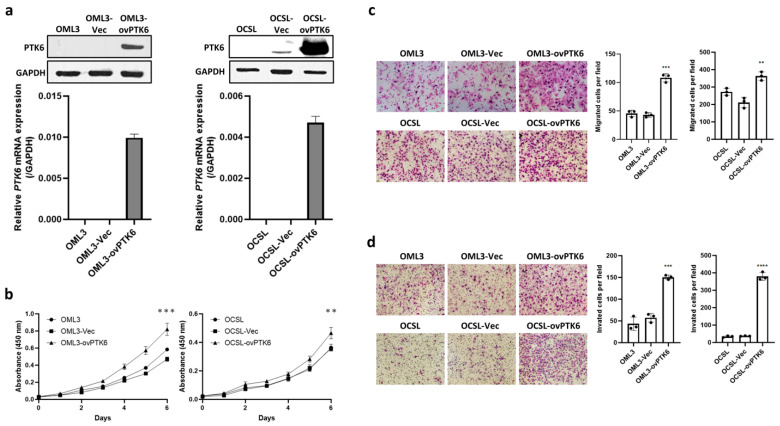
Upregulation of PTK6 facilitated OSCC cell migration and invasion. (**a**) PTK6 overexpression was induced in OML3 and OCSL cell lines by transfecting a PTK6 overexpression plasmid. The mRNA and protein expression were examined through real-time PCR and Western blotting in PTK6 overexpression cells. (**b**) Cell viabilities of control and PTK6 overexpression cells were examined using a WST-1 assay. Absorbance was measured at 450 nm at seven time points (0, 24, 48, 72, 96, 120, and 144 h) to calculate the proliferation rate. (**c**) The migration abilities of control and PTK6 overexpression cells were examined using a transwell assay. Migrated cells were quantified using ImageJ. (**d**) The invasive abilities of control and PTK6 overexpression cells were measured using a transwell Matrigel assay. Invasive cells were quantified using ImageJ. ns means not significant. ** *p* < 0.01; *** *p* < 0.005; **** *p* < 0.001.

**Figure 5 ijms-23-04495-f005:**
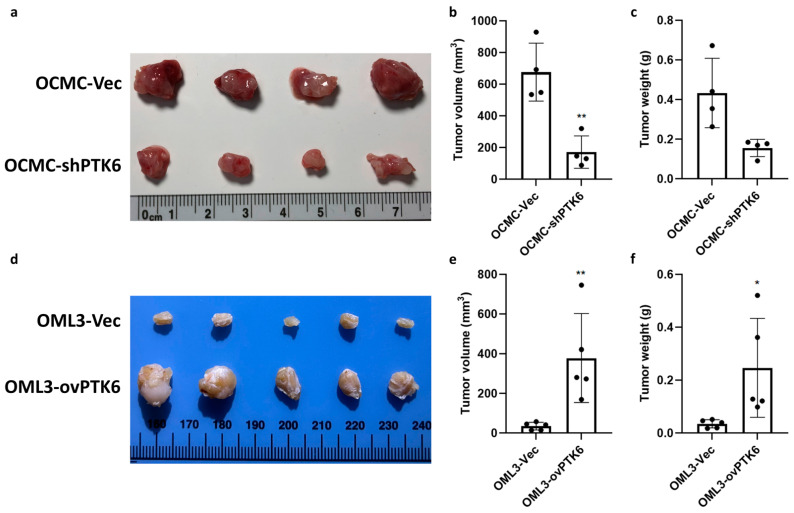
PTK6 promoted tumorigenesis in vivo. (**a**) Images show the sizes of tumors collected from control and PTK6 knockdown xenograft tumor models. (**b**,**c**) Tumor volumes and weights at 4 weeks post injection were measured. (**d**) Images show the sizes of tumors received from control and PTK6 overexpression xenograft tumor models. (**e**,**f**) Tumor volumes and weights at 8 weeks post injection were measured. Tumor volume was calculated using the following formula: TV (mm^3^) = (L × W^2^)/2. * *p* < 0.05. ** *p* < 0.01.

**Figure 6 ijms-23-04495-f006:**
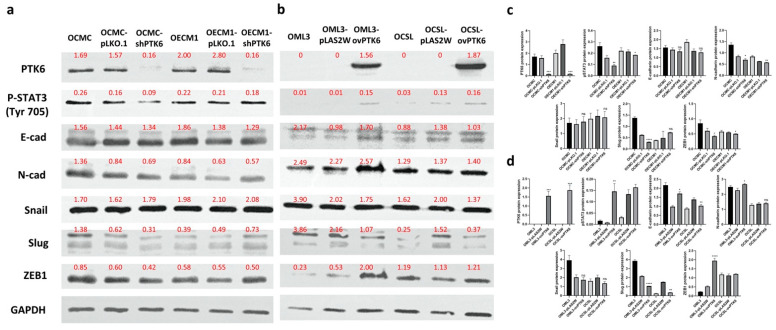
Analysis of candidate genes downstream of PTK6. (**a**,**b**) The protein levels of candidate genes downstream of PTK6 downstream were detected through Western blotting. (**c**,**d**)The intensities of Western blotting data were measured with ImageJ and normalized to GAPDH. * *p* < 0.05; ** *p* < 0.01; *** *p* < 0.005; **** *p* < 0.001; ns: not significant.

## Data Availability

Not applicable.

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
