# Peer review of "Epigenetic Deregulation of Protein Tyrosine Kinase 6 Promotes Carcinogenesis of Oral Squamous Cell Carcinoma"

_ijms, 2022, doi:10.3390/ijms23094495_

Round 1

Reviewer 1 Report

Yi-Ping Hsieh et al. have presented a comprehensive study of Protein Tyrosine Kinase 6 (PTK6) gene epigenetic deregulation in oral squamous cell carcinoma (OSCC), and its role in the OSCC pathogenesis. Their conclusion that the epigenetic deregulation of PTK6 can serve as a biomarker for the early detection of OSCC and a treatment target is well substantiated by the studies performed in vitro and in vivo. The manuscript is finely composed and illustrated, and it can be accepted for publication in present form.

Author Response

Thank you very much for your review of this manuscript.

Reviewer 2 Report

The manuscript „Epigenetic Deregulation of Protein Tyrosine Kinase 6 Promotes Carcinogenesis of Oral Squamous Cell Carcinoma“ lacks some important data that are needed for justification of the title.

The authors worked with native clinical samples (methylation in the PTK6 exon 1), experimental animals (OSCC induction and analyses of methylation and PTK6 expression – mRNA and protein), cell lines and xenografts with overexpressed and silenced PTK6.

In the Introductory part, the authors should offer far more data related to PKM6, and its function with respect to its intracellular localization. Further, they should explain the relationship between PTK6 and STAT3, which was originally discovered back in 2013. Some of these data are presented in Discussion section. Still, PTK6, Snail, ZEB and some other proteins involved in the EMT needs to be mentioned in the introductory part of this manuscript.

Generally, in Material and Methods, the authors should give far more data related to experimental protocols. For example, for silencing experiments, how were they performed, with which transfection reagents, and with which concentration of shPTK6. Catalogue numbers and dilution of antibodies, as well as their species specificity must be provided.

FaDu is for sure not the OSCC cell line; it originates from hypopharynx.

I would like to see the nucleotide sequences S1 and S2, and would like to know their distance from the TSS in both isoforms, the longer one, and the lambdaM5/Alt-PTK6. The primers used for pyrosequencing needs to be positioned accordingly; with respect to the TSS.

Fig 1C and 1D: Were the patients whose saliva was tested for the PTK6 methylation randomized?

With respect to methylation, the authors claim that PTK6 methylation status can be used as a marker, although the ROC analysis indicates specificity and sensitivity of 0.644 and 0.621, respectively. These are not the values which should be attributed to an accurate biomarker. Also, in cell culture experiments, only one HNOK cell line was used, at least according to my understanding. That is not enough for concluding that „that the hypomethylation of PTK6 may be a critical mechanism resulting in the deregulation of PTK6 in OSCC carcinogenesis“. I would like to learn more about the HNOK used.

Suppl. Figure 2C – that WB should be better prepared. It seems to me that the loading control (GAPDH) was not uniformly loaded.

In the legend of Figure 2, the authors state that “PTK6 hypomethylation induces OSCC carcinogenesis…”. That may be, but that was not shown in the stated experiments, because chemicals given act in a non-specific fashion. In other words, they may change the transcription rate of numerous genes, for example, the NRF2 targets.

It is not clear which primers were used for amplifying PTK6 in human and mouse samples. For GAPDH, primers complementary to mouse mRNA are listed. For PTK6, I am able to find the primers complementary to human mRNA. However, the reverse primer is complementary to the part of the longer isoform. Does it mean that the shorter isoform was not amplified with the primers used? What about the mouse sequence? The authors must provide a table with all primers used, the Acc. No. of corresponding sequences deposited in the GeneBank and the size of amplicons expected to be obtained.

I would like to see the WBs related to the Figure 2D, as well as the detailed experimental protocol, including the amount of proteins loaded for WB. How much proteins can be obtained from the mouse oral cavity lesion? Were the samples pooled? How many animals were included in these studies?

For experiments using the WST-1 assay, the authors claim that they measured proliferation and viability. With that test, viability was measured. For proliferation, the authors should have used any of tests based on the BrDU incorporation.

With respect to STAT3, the authors should explore whether total amount of STAT3 changes in experiments performed. In other words, they should explore both, the total STAT3 and phosphorylated STAT3.

Thank you.

Author Response

Thank you very much for your review of this manuscript.

Reviewer’s comments 1

Moderate English changes required

Response: The manuscript have been edited by Wallace Academic Editing.

Reviewer’s comments 2

In the Introductory part, the authors should offer far more data related to PKM6, and its function with respect to its intracellular localization. Further, they should explain the relationship between PTK6 and STAT3, which was originally discovered back in 2013. Some of these data are presented in Discussion section. Still, PTK6, Snail, ZEB and some other proteins involved in the EMT needs to be mentioned in the introductory part of this manuscript.

Response: We have revised the description of the PTK6 intracellular localization, the relationship between PTK6 and STAT3, and the EMT in the introduction. The revised manuscript is as below.

In breast cancer, PTK6 has also been verified to play a critical role in epithelial–mesenchymal transition (EMT). PTK6 cooperates with HER2 and Src to regulate EMT in HER2-positive breast cancer cells. PTK6 suppresses E-cadherin expression and increases mesenchymal markers in nontransformed breast epithelial cells to promote cell migration [22, 23]. Moreover, a high expression of PTK6 in nondividing epithelial cells was observed to suppress cell proliferation and promote cell differentiation and apoptosis [24]. Such dual functions of PTK6 might be associated with its subcellular localization [25]. PTK6 localized at the nucleus phosphorylates the RNA-binding protein Sam68, which suppresses PTK6-induced cell proliferation [26]. By contrast, cytoplasmic PTK6 phosphorylates many intracellular targets to promote oncogenic functions, including p38/MAPK and JAK2/STAT [27, 28]. STAT3 is a substrate of PTK6 in the cytoplasm. In colon cancer, PTK6 interacts with JAK2 and increases STAT3 phosphorylation to enhance stemness and chemoresistance. PTK6 also promotes STAT3 and ERK5 activation to promote cell survival [28].

Reviewer’s comments 3

Generally, in Material and Methods, the authors should give far more data related to experimental protocols. For example, for silencing experiments, how were they performed, with which transfection reagents, and with which concentration of shPTK6. Catalogue numbers and dilution of antibodies, as well as their species specificity must be provided.

Response: We have revised the transfection reagents and information of antibodies in the material and methods. The revised manuscript is as described below.

4.4. PTK6 knockdown and overexpression in cell lines

The shRNA specifically targeting PTK6 and the pLKO.1-emptyT vector were ob-tained from the National RNAi Core of Academia Sinica (clone ID TRCN0000021551, Taipei, Taiwan). 10 µg pLKO.1Ppuro and 10 µg pLKO.1-shPTK6 plasmids were trans-fected into HEK293T cells with Lipofectamine™ 3000 (Invitrogen, Waltham, USA) and produced lentiviruses, respectively. The lentiviruses were used to infect OSCC cell lines (OCMC and OECM1) expressing a high level of PTK6 (multiplicity of infection, MOI=3). The empty vector pLAS2W-Puro was obtained from the National RNAi Core of Academia Sinica. The pLAS2W-PTK6 expression plasmid was synthesized by Protech Technology Enterprise. 10 µg pLAS2W-Puro and 10 µg pLAS2W-PTK6 expression plasmids were transfected into HEK293T cell lines with Lipofectamine™ 3000 and produced lentiviruses, respectively. The lentiviruses were used to infect OSCC cell lines (OML3 and OCSL) expressing a low level of PTK6 (MOI=3). Subsequently, 0.5 µg/mL of puromycin was used to select vector control and PTK6 knockdown and overexpression cells. After treatment with puromycin for 3 days, real-time reverse transcription polymerase chain reaction and Western blotting were used to analyze PTK6 mRNA and protein expression. The OSCC PTK6 knockdown or overexpression cells were used to analyze cell functions.

4.6. Western Blotting

Western blotting was performed to investigate the expression of genes downstream of PTK6. Total cellular proteins were extracted using RIPA lysis buffer and a protease inhibitor cocktail (Sigma-Aldrich, Saint Louis, USA). Then, the protein concentration was quantified using the Bio-Rad protein assay kit (Bio-Rad, California, USA). Protein lysates were separated through sodium dodecyl sulfate–polyacrylamide gel electrophoresis and electroblotted onto polyvinylidene fluoride membranes (Millipore, Massachusetts, USA). Then, the membranes were incubated with a primary antibody, followed by a horse-radish peroxidase (HRP)-conjugated secondary antibody. The protein bands were de-tected using a chemiluminescent HRP substrate (Millipore, Massachusetts, USA). The relative expression level of proteins was measured using ImageJ software. GAPDH was used as the internal control. The following antibodies were used: human PTK6 (1:1000 dilution)( sc-166171, Santa Cruz Biotechnology, Texas, USA), human p-STAT3 (1:1000 dilution) (9131,Cell signaling, Massachusetts, USA), human E-cadherin (1:1000 dilution)( 20874-1-AP, Proteintech, Illinois, USA), human N-cadherin (1:1000 dilution) (22018-1-AP, Proteintech, Illinois, USA), human Snail (1:1000 dilution)(A5243, Abclonal, Massachusetts, USA), human Slug (1:1000 dilution)(12129-1-AP, Proteintech, Illinois, USA), human ZEB1(1:1000 dilution)(A16981, Abclonal, Massachusetts, USA) and human GAPDH (1:1000 dilution)(G8795, Sigma-Aldrich, Saint Louis, USA).

Reviewer’s comments 4

FaDu is for sure not the OSCC cell line; it originates from hypopharynx.

Response: We used HNOK, OCMC, OECM1, OC2, OML3, OCSL, SAS, FaDu, and SCC-25 to investigate the mRNA and protein expression of PTK6 in cell lines. We know FaDu is originates from hypopharyngeal tumor. In this research, we used FaDu as reference cell line. The reasons we used FaDu are according the reference papers as  below.

(1) V. S. Chaudhari, B. Gawali, P. Saha, V. G. M. Naidu, U. S. Murty and S. Banerjee. Quercetin and piperine enriched nanostructured lipid carriers (NLCs) to improve apoptosis in oral squamous cellular carcinoma (FaDu cells) with improved biodistribution profile. Eur J Pharmacol 2021, 909, 174400. 10.1016/j.ejphar.2021.174400

(2) C. Z. Lin, Z. Q. Liu, W. K. Zhou, T. Ji and W. Cao. Effect of the regulator of G-protein signaling 2 on the proliferation and invasion of oral squamous cell carcinoma cells and its molecular mechanism. Hua Xi Kou Qiang Yi Xue Za Zhi 2021, 39, 320-327. 10.7518/hxkq.2021.03.012

Reviewer’s comments 5

I would like to see the nucleotide sequences S1 and S2, and would like to know their distance from the TSS in both isoforms, the longer one, and the lambdaM5/Alt-PTK6. The primers used for pyrosequencing needs to be positioned accordingly; with respect to the TSS.

Response: According the NCBI gene data base, There are the same TSS and exon 1 of the both isoforms of PTK6 (as figure below). We have also revised the material and method and add the S1 and S2 nucleotide sequences and location as described below.

PTK6 S1 nucleotide sequences (+101~+151): TYGGAYGGAYGAGGAGTTGAG-TTTTYGYGYGGGGGAYGTTTTTTAYGTGG. PTK6 S2 nucleotide sequences (+243~+293): TATTTGGTYGAGAGGGAGAYGGTGGAGTYGGAATYGTGYG TGTTTTAGGT. The primer sequences were used in PCR and pyrosequencing was shown in supplementary files.

Reviewer’s comments 6

Fig 1C and 1D: Were the patients whose saliva was tested for the PTK6 methylation randomized?

Response: Fig 1C is the DNA methylation of PTK6 in tumor tissues and their corresponding normal tissues from the same patient who were willing to provide contralateral normal oral mucosa samples. Fig 1D is the DNA methylation of PTK6 in saliva. The saliva samples were collected before the patients who were arranged for therapy. 10 normal and 10 OSCC patients who were willing to provide saliva samples were recruited.

Reviewer’s comments 7

With respect to methylation, the authors claim that PTK6 methylation status can be used as a marker, although the ROC analysis indicates specificity and sensitivity of 0.644 and 0.621, respectively. These are not the values which should be attributed to an accurate biomarker. Also, in cell culture experiments, only one HNOK cell line was used, at least according to my understanding. That is not enough for concluding that „that the hypomethylation of PTK6 may be a critical mechanism resulting in the deregulation of PTK6 in OSCC carcinogenesis“. I would like to learn more about the HNOK used.

Response: We performed the ROC analysis of the TCGA database which includes all the head and neck cancer. However, we only focused on oral cancer in this research. We confirmed the PTK6 methylation of specific region 1 or specific region 2 was significantly decreased during OSCC development in clinical samples (Figure 1-b. P < 0.005) and animal models (Figure 2-b. P < 0.05). The results support the epigenetic deregulation of PTK6 to be a biomarker for OSCC detection. HNOK and OSCC cell lines were the cell lines model to validated the clinical results in vitro.

Reviewer’s comments 8

Suppl. Figure 2C – that WB should be better prepared. It seems to me that the loading control (GAPDH) was not uniformly loaded.

Response: We have revised the manuscript and the raw data of western blotting were listed in the supplementary Figure.

Reviewer’s comments 9

In the legend of Figure 2, the authors state that “PTK6 hypomethylation induces OSCC carcinogenesis…”. That may be, but that was not shown in the stated experiments, because chemicals given act in a non-specific fashion. In other words, they may change the transcription rate of numerous genes, for example, the NRF2 targets.

Response: 4-NQO and arecoline treatment was established to OSCC carcinogenesis research in the animal model. After treatment, the normal oral mucosa gradual developed to OPMDs and eventually OSCC. The methylation level of PTK6 decreased with the 4-NQO and arecoline treatment was validated during carcinogenesis of OSCC in this animal model. The mRNA and protein expression of PTK6 increased with time in the mouse tongue samples. Therefore, PTK6 hypomethylation is the critical mechanism relative to the mRNA expression during OSCC carcinogenesis.

Reviewer’s comments 10

It is not clear which primers were used for amplifying PTK6 in human and mouse samples. For GAPDH, primers complementary to mouse mRNA are listed. For PTK6, I am able to find the primers complementary to human mRNA. However, the reverse primer is complementary to the part of the longer isoform. Does it mean that the shorter isoform was not amplified with the primers used? What about the mouse sequence? The authors must provide a table with all primers used, the Acc. No. of corresponding sequences deposited in the GeneBank and the size of amplicons expected to be obtained.

Response: We had revised the manuscript and the primers were used for amplifying PTK6 in human and mouse samples were described. The figures below were the blast results of our primers

Reviewer’s comments 11

I would like to see the WBs related to the Figure 2D, as well as the detailed experimental protocol, including the amount of proteins loaded for WB. How much proteins can be obtained from the mouse oral cavity lesion? Were the samples pooled? How many animals were included in these studies?

Response: We had revised the manuscript and the experimental protocol as described below.

Animal experiments were approved by the Animal Care and Use Committee (IACUC) of National Cheng Kung University. Male C57BL/6J mice raised on the Tainan campus of the National Institutes of Health were used in this study. The mice were randomly divided into the control and experimental groups. The control group was given normal drinking water. The experimental group was given drinking water containing 4-NQO (200 μg/mL; Sigma-Aldrich, Saint Louis, USA) and arecoline (200 μg/mL; TCI, Tokyo, Japan) to induce the development of OSCC. The mice in the control and experimental groups were sacrificed at 8, 16, 20, and 29 weeks and the different stages of OSCC development were determined. There are 8 weeks group included 3 control and 7 4-NQO and arecoline treated mice in 8 weeks group; 3 control and 4 4-NQO and arecoline treated mice in 16 weeks group; 3 control and 6 4-NQO and arecoline treated mice in 20 weeks group; and 3 control and 4 4-NQO and arecoline treated mice in 29 weeks group. About 3x12 mm tissue was cut from each mouse, and about 150 µg of protein lysate was obtained. 50 µg total protein lysate was used for western blotting analysis. DNA and RNA were extracted from the tongue tissues of the mice to examine changes in PTK6 expression in the different stages of OSCC development.

Reviewer’s comments 12

For experiments using the WST-1 assay, the authors claim that they measured proliferation and viability. With that test, viability was measured. For proliferation, the authors should have used any of tests based on the BrDU incorporation.

Response: WST-1 proliferation assays have become available for analyzing the number of viable cells by the cleavage of tetrazolium salts added to the culture medium. The tetrazolium salts are cleaved to formazan by cellular enzymes. An expansion in the number of viable cells results in an increase in the overall activity of mitochondrial dehydrogenases in the sample, which in turn increases the amount of formazan dye formed. Quantification of the formazan dye produced by metabolically active cells can be done using a scanning multiwell spectrophometer. As the description above, we use WST-1 assay for measuring proliferation and viability.

Reviewer’s comments 13

With respect to STAT3, the authors should explore whether total amount of STAT3 changes in experiments performed. In other words, they should explore both, the total STAT3 and phosphorylated STAT3.

Response: PTK6 is a kinase that promotes STAT3 phosphorylation without affects STAT3 expression. Therefore, we used phosphorylated STAT3 specific antibody to measure the STAT3 active form. The paper below is the reference that we accounting for.

  1. Liu, Z. Pan, Q. Chen, Z. Chen, W. Liu, L. Wu, M. Jiang, W. Lin, Y. Zhang, W. Lin, R. Zhou and L. Zhao. Pharmacological targeting PTK6 inhibits the JAK2/STAT3 sustained stemness and reverses chemoresistance of colorectal cancer. J Exp Clin Cancer Res 2021, 40, 297. 10.1186/s13046-021-02059-6

Reviewer 3 Report

In this study, the authors found that in oral squamous cell carcinoma, the methylation level of the first exon of PTK6 decreased, and the expression level increased accordingly. They also suggested that PTK6-induced promotion of proliferation, migration and invasion is associated with elevated STAT3 phosphorylation levels and ZEB1 levels in oral cancer cells. Based on these results, the authors argue that epigenetic deregulation of PTK6 can be a therapeutic target for OSCC as well as a biomarker for early OSCC detection.

I agree with the authors' claim that PTK6 hypomethylation may be a biomarker for OSCC development. On the other hand, the correlation between PTK6 methylation regulation and gene expression fluctuation, carcinogenesis or metastasis may require further investigation.

  1. In this study, it has been shown that the methylation levels of S1 and S2 present in the first exon of the PTK6 gene decrease with carcinogenesis. These results imply that expression levels are low in normal cells because S1/S2 are methylated, and that a decrease in those methylation levels is associated with increased expression. However, no data have been shown that S1/S2 methylation levels are associated with the regulation of gene expression. Even if they cannot provide data to prove directly, they must at least provide literature suggesting a link between intragenic CpG methylation and expression regulation.
  2. In Figure 3, downregulation of PTK6 did not affect cell proliferation in OCMC. However, in Figure 5, the size of the OCMC xenograft tumor with PTK6 knockdown is clearly reduced. One of the causes of the discrepancy in these data is considered to be cell death. Data on the induction of cell death associated with decreased PTK6 levels should be presented.
  3. Both STAT3 phosphorylation and ZEB1 levels appear to be elevated when PTK6 is knocked down on OECM1. These data suggest that PTK6 does not always positively control these levels. The authors need to show how they think about that.
  4. Since the results of Western blotting are subtle, it is necessary to perform a significant difference test using multiple data and graph it.

Author Response

Thank you very much for your review of this manuscript.

Reviewer’s comments 1

Moderate English changes required

Response: The manuscript have been edited by Wallace Academic Editing.

Reviewer’s comments 2

In this study, it has been shown that the methylation levels of S1 and S2 present in the first exon of the PTK6 gene decrease with carcinogenesis. These results imply that expression levels are low in normal cells because S1/S2 are methylated, and that a decrease in those methylation levels is associated with increased expression. However, no data have been shown that S1/S2 methylation levels are associated with the regulation of gene expression. Even if they cannot provide data to prove directly, they must at least provide literature suggesting a link between intragenic CpG methylation and expression regulation.

Response: The manuscript has been revised. The reference about the correlation between first exons methylation and gene expression had revised in the discussion as below. The hypomethylation of PTK6 first exons methylation and PTK6 expression increase were validated as data in Figure 2 and supplementary Figure 2.

Previous studies indicated the methylation densities in promoters and first exons were negatively correlated with the corresponding gene expression level [36]. Two specific loci on the PTK6 exon 1 and analyzed the methylation status of CG sites were identified in this research. The PTK6 methylation of specific region 1 or specific region 2 was gradually decreased during the carcinogenesis of OSCC were also validated.

Reviewers’ comments 3

In Figure 3, downregulation of PTK6 did not affect cell proliferation in OCMC. However, in Figure 5, the size of the OCMC xenograft tumor with PTK6 knockdown is clearly reduced. One of the causes of the discrepancy in these data is considered to be cell death. Data on the induction of cell death associated with decreased PTK6 levels should be presented.

Response: Tumor microenvironment is complex, there are many factors that influence tumor growth in vivo. Tumor invasiveness, hypoxia, angiogenesis, and apoptosis may be the critical factor that affect tumor growth in vivo. In breast cancer research, downregulation of PTK6 has been identified to promote apoptosis by inducing Bim expression and p38 MAPK activation, especially in therapy-resistant breast cancer cells (references as below). The correlation between PTK6 and apoptosis is needed to be explored in our future research.

  1. Ito, S. H. Park, I. Katsyv, W. Zhang, C. De Angelis, R. Schiff and H. Y. Irie. PTK6 regulates growth and survival of endocrine therapy-resistant ER+ breast cancer cells. NPJ Breast Cancer 2017, 3, 45. 10.1038/s41523-017-0047-1
  2. H. Park, K. Ito, W. Olcott, I. Katsyv, G. Halstead-Nussloch and H. Y. Irie. PTK6 inhibition promotes apoptosis of Lapatinib-resistant Her2(+) breast cancer cells by inducing Bim. Breast Cancer Res 2015, 17, 86. 10.1186/s13058-015-0594-z

Reviewers’ comments 4

Both STAT3 phosphorylation and ZEB1 levels appear to be elevated when PTK6 is knocked down on OECM1. These data suggest that PTK6 does not always positively control these levels. The authors need to show how they think about that.

Response: We revised the manuscript and the intensity of western blotting data by Image J and normalizing to the GAPDH were in listed Figure 6. The expression level of STAT3 phosphorylation and ZEB1 was decreased in the PTK6 knockdown OECM1 cell line. The correlation between PTK6, STAT3 phosphorylation, and ZEB1 in OSCC were validated. However, the signaling transduction is complex. The detail dissecting of the mechanism should be done in our future research.

Reviewers’ comments 5

Since the results of Western blotting are subtle, it is necessary to perform a significant difference test using multiple data and graph it.

Response: We revised the manuscript and the intensity of western blotting data by Image J and normalizing to the GAPDH were in listed Figure 6.

Round 2

Reviewer 2 Report

I thank to the authors for offering some additional clarifications and for their efforts for improving this manuscript.

The authors did not address all my concerns and requests, and some explanation given are simply not acceptable (comment #4: FaDu as a reference cell line?, comment #2: no WB images with respect to Fig. 2D). Now, when I see the explanation offered, I wonder why the authors did not perform IHC, for mouse oral cavity lesions (their tongues, specifically).

No referal sequences accession numbers.

With respect to #7: „AUC is an effective way to summarize the overall diagnostic accuracy of the test. It takes values from 0 to 1, where a value of 0 indicates a perfectly inaccurate test and a value of 1 reflects a perfectly accurate test. AUC can be computed using the trapezoidal rule.3 In general, an AUC of 0.5 suggests no discrimination (i.e., ability to diagnose patients with and without the disease or condition based on the test), 0.7 to 0.8 is considered acceptable, 0.8 to 0.9 is considered excellent, and more than 0.9 is considered outstanding.“ (PMID#; 20736804 over 1000 citation).

What is bellow 0.7 is not acceptable. Cannot be diagnostic marker („Supplementary Figure 1. PTK6 hypomethylation can serve as a diagnostic marker of HNSC. The TCGA data were used to validate the diagnostic value of PTK6 hypomethylation in HNSC by plotting the ROC curve.“).

An alternate name for PTK6 is BRC. Its role was originally explored in OSCC in 2004 (PMID: 15509496). These findings are not discussed here. With respect to laryngeal cancer (HNSCC), „LSCC patients with low expression of PTK6 had reduced DFS and OS, and patients with medium PTK6 expression had worse DFS and OS than those with high expression of PTK6, suggesting that PTK6 loss from non-cancerous tissue could be an early event in transformation“.

Thank you.

Author Response

Thank you very much for your review of this manuscript.

Reviewer’s comments 1

The authors did not address all my concerns and requests, and some explanation given are simply not acceptable (comment #4: FaDu as a reference cell line?, comment #2: no WB images with respect to Fig. 2D). Now, when I see the explanation offered, I wonder why the authors did not perform IHC, for mouse oral cavity lesions (their tongues, specifically).

Response: In preventing of confusion, we revised the supplementary Figure 3. The FaDu cell line was removed, and the statistical analysis was re-done.

  We have also revised the supplementary and the PTK6 protein expression in the animal model of western blot images and IHC stain were shown in Supplementary Figure 3, and Supplementary Figure 4.

Reviewer’s comments 2

No referal sequences accession numbers.

Response: We have revised the Supplementary Table 1 and the referral sequences accession numbers were shown.

Reviewer’s comments 3

With respect to #7: „AUC is an effective way to summarize the overall diagnostic accuracy of the test. It takes values from 0 to 1, where a value of 0 indicates a perfectly inaccurate test and a value of 1 reflects a perfectly accurate test. AUC can be computed using the trapezoidal rule.3 In general, an AUC of 0.5 suggests no discrimination (i.e., ability to diagnose patients with and without the disease or condition based on the test), 0.7 to 0.8 is considered acceptable, 0.8 to 0.9 is considered excellent, and more than 0.9 is considered outstanding.“ (PMID#; 20736804 over 1000 citation).

What is bellow 0.7 is not acceptable. Cannot be diagnostic marker („Supplementary Figure 1. PTK6 hypomethylation can serve as a diagnostic marker of HNSC. The TCGA data were used to validate the diagnostic value of PTK6 hypomethylation in HNSC by plotting the ROC curve.“).

Response: We performed the ROC analysis of the TCGA database which includes all the head and neck cancer. We agree PTK6 hypomethylation may not be a significant diagnostic marker in head and neck cancer according to the experiment data of TCGA. However, we only focused on oral squamous cell carcinoma (OSCC) in this research and the pyrosequencing analysis of PTK6 S1, and S2 loci were validated in our research. In our experiments, the PTK6 methylation of S1, and S2 loci were significantly decreased by pyrosequencing analysis during OSCC development in clinical samples (Figure 1-b. P < 0.005) and animal models (Figure 2-b. P < 0.05). These results support the epigenetic deregulation of PTK6 has the potential to be a biomarker for OSCC detection.

  We have also revised the figure legend of Supplementary Figure 1 as described below.

Supplementary Figure 1. The TCGA data were used to show the diagnostic value of PTK6 hypomethylation in HNSC by plotting the ROC curve.

Reviewer’s comments 4

An alternate name for PTK6 is BRC. Its role was originally explored in OSCC in 2004 (PMID: 15509496). These findings are not discussed here. With respect to laryngeal cancer (HNSCC), „LSCC patients with low expression of PTK6 had reduced DFS and OS, and patients with medium PTK6 expression had worse DFS and OS than those with high expression of PTK6, suggesting that PTK6 loss from non-cancerous tissue could be an early event in transformation“.

Response: We have revised the discussion as below.

PTK6 upregulation has been identified in multiple cancer types, and associated with poor patient prognosis [18, 33-35]. In previously research of investigated expression of PTK6 in human oral squamous cell carcinomas and normal oral epithelium (NOE). The results show NOE express higher levels of PTK6 compared with OSCC cells [36]. However, mechanisms underlying PTK6 dysregulation in OSCC remain unclear. Previous studies also indicated the methylation densities in promoters and first exons were negatively correlated with the corresponding gene expression level [37]. Two specific loci on the PTK6 exon 1 and analyzed the methylation status of CG sites were identified in this research. The PTK6 methylation of specific region 1 or specific region 2 was gradually decreased during the carcinogenesis of OSCC were also validated. The results were confirmed in the animal model treated with 4-NQO and arecoline that exhibited a decrease in the methylation level of PTK6 with time. Subsequently, the mRNA and protein expression of PTK6 increased following treatment with 4-NQO and arecoline. The methylation of PTK6 was decreased, and the mRNA and protein expression of PTK6 was increased in the OSCC cells compared with HNOKs. Therefore, the methylation of PTK6 was negatively correlated with its transcription level. We determined that PTK6 hypomethylation increased the PTK6 transcription level in OSCC progression. The main risk factors of OSCC in Taiwan is betel quid chewing [38]. However, use of tobacco, alcohol and HPV infection are the main etiological risk factor for OSCC in the USA [39-41]. Different risk factors induce OSCC carcinogenesis may involve in different mechanism. Therefore, the PTK6 expression in OSCC is worth to explore detailly in different country.

Round 3

Reviewer 2 Report

I thank authors for further clarifications and data offered.

Unfortunately, I still see serious issues: The images of IHC offered are of a very poor quality; b) WBs: it seems to me that there is a striking similarity between some lines on Figure 6: for example, ZEB 1 and N-cadherin. What also concerns me is the varying intensity of referral bands of the protein standard (on some images 72 kDa is predominant, but not on all).

With respect to sequences listed, I can only find a partial complementarity with the S2 region, but I do have a problem with the S1 region; cannot find it on a human sequence: I do not see the stretch of six "T"s, presented in the manuscript ((+101~+151): TYGGAYGGAYGAGGAGTTGAGTTTTYGYGYGGGGGAYGTTTTTTAYGTGG)? 

CTGGGTGTGGCTGCTTCCCGGCCTGTCATGAGGAAGTGGGACGGCCCGCCTGCC +1ACGCCCAGCTCTGGGTGCTCCAGCTGGGCCACAGCCTGGTCCTGCCGCTGCGCCCGCCCGCCATGGTGTCCCGGGACCAGGCTCACCTGGGCCCCAAGTATGTGGGCCTCTGGGACTTCAAGTCCCGGACGGACGAGGAGCTGAGCTTCCGCGCGGGGGACGTCTTCCACGTGGCCAGGAAGGAGGAGCAGTGGTGGTGGGCCACGCTGCTGGACGAGGCGGGTGGGGCCGTGGCCCAGGGCTATGTGCCCCACAACTACCTGGCCGAGAGGGAGACGGTGGAGTCGGAACC

red-"mute"

yellow-non-translated part of the first exon

blue: coding part of the first exon; ATG- first coding exon

I have also checked the referral PTK6 mRNA:

NM_005975.4 Homo sapiens protein tyrosine kinase 6 (PTK6), transcript variant 1, mRNA

ACGCCCAGCTCTGGGTGCTCCAGCTGGGCCACAGCCTGGTCCTGCCGCTGCGCCCGCCCGCCATGGTGTCCCGGGACCAGGCTCACCTGGGCCCCAAGTATGTGGGCCTCTGGGACTTCAAGTCCCGGACGGACGAGGAGCTGAGCTTCCGCGCGGGGGACGTCTTCCACGTGGCCAGGAAGGAGGAGCAGTGGTGGTGGGCCACGCTGCTGGACGAGGCGGGTGGGGCCGTGGCCCAGGGCTATGTGCCCCACAACTACCTGGCCGAGAGGGAGACGGTGGAGTCGGAACC

Thank you.

Author Response

Thank you very much for your review of this manuscript.

Reviewer’s comments 1

Unfortunately, I still see serious issues: The images of IHC offered are of a very poor quality;

Response: We revised the IHC image in Supplementary Figure 4.

Reviewer’s comments 2

  1. b) WBs: it seems to me that there is a striking similarity between some lines on Figure 6: for example, ZEB 1 and N-cadherin.

Response: Raw data of western blots are shown in Supplementary. Images show different molecular weights of ZEB 1 and N-cadherin. The bands of ZEB 1 and N-cadherin are different.

Reviewer’s comments 3

What also concerns me is the varying intensity of referral bands of the protein standard (on some images 72 kDa is predominant, but not on all).

Response: Western blots are captured under CoolSnap with immobilon western chemiluminescent HRP substrate (Millipore, WBKLS0500) in our experiments. However, protein markers only can be visualized clearly under white light and is fading under 425 nm filter fluorescence detection. The protein marker in western blots of Slug is predominate due to the loading quantity was more than other experiments. The protein markers over the right and left showed different protein markers loading in 3 experiments of Slug.

Reviewer’s comments 4

With respect to sequences listed, I can only find a partial complementarity with the S2 region, but I do have a problem with the S1 region; cannot find it on a human sequence: I do not see the stretch of six "T"s, presented in the manuscript ((+101~+151): TYGGAYGGAYGAGGAGTTGAGTTTTYGYGYGGGGGAYGTTTTTTAYGTGG)? 

Response: ”TYGGAYGGAYGAGGAGTTGAGTTTTYGYGYGGGGGAYGTTTTTTAYGTGG” was designed for pyrosequence to analyze the methylation level. We performed the bisulfite conversion before pyrosequencing, unmethylated cytosines were convert to thymine after bisulfite treatment and PCR. The six "T"s were designed for detection of “TCTTCC”

The original sequences detected was listed as below:

Human PTK6 S1: CCGGACGGACGAGGAGCTGAGCTTCCGCGCGGGGGACGTCTTCCACGTGGCC

Human PTK6 S2:

TACCTGGCCGAGAGGGAGACGGTGGAGTCGGAACCGTGCG

CTGGGTGTGGCTGCTTCCCGGCCTGTCATGAGGAAGTGGGACGGCCCGCCTGCC +1ACGCCCAGCTCTGGGTGCTCCAGCTGGGCCACAGCCTGGTCCTGCCGCTGCGCCCGCCCGCCATGGTGTCCCGGGACCAGGCTCACCTGGGCCCCAAGTATGTGGGCCTCTGGGACTTCAAGTCCCGGACGGACGAGGAGCTGAGCTTCCGCGCGGGGGACGTCTTCCACGTGGCCAGGAAGGAGGAGCAGTGGTGGTGGGCCACGCTGCTGGACGAGGCGGGTGGGGCCGTGGCCCAGGGCTATGTGCCCCACAACTACCTGGCCGAGAGGGAGACGGTGGAGTCGGAACCGTGCG

(Y means possible methylation sites; orange T means unmethylated cytosines were convert to thymine after bisulfite treatment and PCR)

We revised the material and method and Supplementary Table 1.
